# The Interaction between Viral and Environmental Risk Factors in the Pathogenesis of Multiple Sclerosis

**DOI:** 10.3390/ijms20020303

**Published:** 2019-01-14

**Authors:** Rachael Eugenie Tarlinton, Timur Khaibullin, Evgenii Granatov, Ekaterina Martynova, Albert Rizvanov, Svetlana Khaiboullina

**Affiliations:** 1School of Veterinary Medicine and Science, University of Nottingham, LE12 9RH Nottingham, UK; rachael.tarlinton@nottingham.ac.uk; 2Republican Clinical Neurological Center, Republic of Tatarstan, Kazan 420021, Russia; timuur@gmail.com (T.K.); evgranatov@gmail.com (E.G.); 3Department of Gene and Cell Technology, Institute of Fundamental Medicine and Biology, Republic of Tatarstan, Kazan 420021, Russia; ignietferro.venivedivici@gmail.com (E.M.); albert.rizvanov@kpfu.ru (A.R.); 4Department of Microbiology and Immunology, University of Nevada, Reno, NV 89557, USA

**Keywords:** cytotoxic T lymphocytes (CTL), multiple sclerosis, central nervous system (CNS), Epstein Barr Virus (EBV), HERV

## Abstract

Multiple sclerosis (MS) is a chronic debilitating inflammatory disease of unknown ethology targeting the central nervous system (CNS). MS has a polysymptomatic onset and is usually first diagnosed between the ages of 20–40 years. The pathology of the disease is characterized by immune mediated demyelination in the CNS. Although there is no clinical finding unique to MS, characteristic symptoms include sensory symptoms visual and motor impairment. No definitive trigger for the development of MS has been identified but large-scale population studies have described several epidemiological risk factors for the disease. This list is a confusing one including latitude, vitamin D (vitD) levels, genetics, infection with Epstein Barr Virus (EBV) and endogenous retrovirus (ERV) reactivation. This review will look at the evidence for each of these and the potential links between these disparate risk factors and the known molecular disease pathogenesis to describe potential hypotheses for the triggering of MS pathology.

## 1. Introduction

### 1.1. MS Pathology and Clinical Disease

MS pathology is characterized by multiple lesions within the CNS [1]. The pathogenesis of brain lesions remains largely unknown; however, neurodegeneration due to inflammation and immune reaction towards the brain cells is believed to play the central role. Interestingly, brain plaques in MS are predominantly found within the white matter around the lateral ventricles and optic nerve [2,3]. Recently, the presence of the brain plaques in the grey matter was demonstrated by Calabrese et al. [4]. These lesions could be detected early and correlate with the disease severity [5]. Lesions in periventricular locations along the 4th ventricle, midbrain and cerebellar peduncle are also characteristic for MS, though they are less prevalent [6,7]. The close proximity of lesions to the ventricles containing the choroid plexus, a structure of the blood-cerebrospinal fluid (CSF) barrier, suggests a connection between peripheral circulation and brain tissue pathology (Figure 1).

Several factors (sunlight exposure, serum vitamin D (vitD) level, viral infection and genetic factors) trigger development of autoreactive T lymphocytes in the periphery. T lymphocytes targeting myelin cross the blood-brain barrier (BBB) and reach the neurons in CNS. Within the brain, T lymphocytes trigger inflammation and myelin degradation. Consequently, chronic inflammation and reduced myelin will affect the neuron function, leading to clinical symptoms of Multiple Sclerosis.

This assumption is supported by the finding that activated mononuclear cells, such as lymphocytes, macrophages and dendritic cells, occur within the brain plaques near the BBB [8,9,10,11]. It appears that MS lymphocytes have a higher ability to cross the BBB when compared to cells from healthy controls [12]. Leukocyte migration is tightly regulated and involves interaction between cell adhesion molecules (CAMs) on BBB endothelial cells and their ligands on the white blood cells [13,14,15]. The very late antigen 4 (VLA-4) ligand interaction with VCAM appears essential for activated T-cell migration across the BBB in MS, as VLA-4^+^ leukocyte infiltration was demonstrated in MS lesions [16]. Additionally, the therapeutic effect of interferon (IFN) was linked to decreased expression of this ligand on circulating leukocytes in MS [17]. Therefore, targeting VLA-4 was suggested for treatment of MS, which was shown to be effective in RRMS [18]. Decreased counts of circulating CD4^+^, CD8^+^ and CD19^+^ lymphocytes together with reduced expression of VLA-4 ligand on activated leukocytes was also linked to therapeutic efficacy of natalizumab, a humanized monoclonal antibody binding to VLA-4 [19,20,21]. Interestingly, leukocyte migration can be inhibited by IFN-β and tissue inhibitor of metalloprotease 1 (TIMP-1) [22,23], suggesting that this process is regulated by immune mediators and requires metalloprotease. This assumption was further supported by Prat et al. demonstrating that antibody to CCL2, a potent mononuclear leukocyte chemoattractant [24], has significantly reduced leukocyte migration in MS [25].

Currently, the leading role of Th17 and Th1 lymphocytes in pathogenesis of the MS brain plaques is well recognized [26,27]. High numbers of Th1 lymphocytes, expressing IL-12 and IFN-γ, are commonly found in the brains of experimental autoimmune encephalomyelitis (EAE) mice, the animal model of MS [28]. Also, elevated numbers of CD8^+^ and CD4^+^ lymphocytes producing IL-17 have been found in active brain lesions in MS cases [29]. Th17 cells can readily cross the BBB, as they produce cytokines, chemokines and express receptors that compromise tissue barrier permeability [30,31,32]. Based on the large body of evidence supporting the role of Th1 and Th17 lymphocytes in MS pathogenesis, it has now been suggested that the Th17 response is more relevant in the early stages of the disease, while the Th1 lymphocytes are important later, supporting local inflammation [33]. B cells are also involved in MS pathogenesis with studies demonstrating increased B cell counts [34] and the presence of oligoclonal antibody complex bands in MS CSF [35]. The most compelling evidence for the role of B cells in MS pathogenesis comes from clinical trials using the CD20 monoclonal antibodies rituximab, ocrelizumab and ofatumumab [36,37,38], which deplete the B cell population. Collected data from these trials demonstrated that treatment with rituximab reduced the number of new lesions and proportion of patients experiencing relapses [39].

Autoreactive lymphocytes target basic myelin protein, which is a component of the myelin sheets that sheath neurons. Oligodendrocyte apoptosis is detected at the site of demyelinization in MS [40] resulting in neuroglial activation [41,42]. Myelin reactive lymphocytes are found in the circulation and CSF of MS cases as compared to healthy subjects [43,44]. The role of myelin-specific leukocytes in MS pathogenesis was confirmed using EAE mice, where injection of myelin-specific CD8^+^ cytotoxic lymphocytes led to severe CNS autoimmunity in animals [45]. This immune response is, however, polyclonal with no clear overlap in antibody or T cell clones between individual MS patients or different studies, nor is it clear whether structural changes to the protein itself are triggers of the immune response that typifies MS or responses to it [46,47,48].

Clinically, several forms of MS are recognized: relapsing remitting MS (RRMS), secondary progressing MS (SPMS), primary progressing MS (PPMS) and progressive relapsing (PRMS) (Figure 2). In 80–85% of cases, the onset of RRMS is characterized by episodes of neurological disability and recovery [49]. As the RRMS progresses, 60–70% of cases will gradually worsen with a steady progression of symptoms [50]. This pattern of the disease is referred to as SPMS. A small group of patients (approximately 10%) will develop PPMS, which is characterized by steady progression of the neurological symptoms without periods of recovery [51,52].

Each form of MS differs in the frequency of exacerbations and duration of the remission. Interestingly, it appears that periventricular lesions are more often found in progressive MS as compared to the relapsing form of the disease [54]. However, the number of lesions and their localization in the spinal cord do not correlate with the form of the disease or disability. Brain lesion localization and their number are related to clinical symptoms and clinical presentation, where the number of demyelination foci increases each time the patient experiences a clinical relapse [55]. However, it appears that brain inflammation does continue in remission with the number of brain plaques increasing gradually during disease remission. Clinical symptoms of MS vary depending on the localization of the brain lesions, as they affect structures connected specifically to those parts of the CNS [56]. Spinal cord involvement presents with sensory loss or motor weakness in the body, while damage to the brainstem affects sensation or weakness of the face and diplopia. When inflammation is localized in the optic nerve, signs such as blurry vision, ocular pain and visual loss are typically described [57].

### 1.2. Gene-Environment Interaction

Various studies suggest a genetic predisposition to MS. There is a clear difference in risk of developing MS between different ethnic groups. Although diagnosed worldwide, the highest MS incidence rate is registered in Europe [58]. The highest prevalence (≥200/100,000) is in Scotland, Northern Ireland and within certain populations in Scandinavia and Sicily [59,60,61]. MS prevalence in the British Isles ranged from 96/100,000 to more than 200/100,000, with the highest rates in Scotland and Northern Ireland (7.2 to 12.2 per 100,000) [62,63,64]. It appears that populations of Northern European descent are at higher risk of developing MS. Eight-fold lower rates of MS have been found in Asian and African populations in Norway when compared to the indigenous Sami [65,66]. Similarly, in Malta, Northern European immigrants had a 10-times higher MS prevalence than that of Maltese-born individuals [67]. These data also suggest a genetic predisposition to MS independent of latitude effects.

There is also a wealth of evidence supporting a higher frequency of MS diagnosis in siblings [68,69,70] and individuals closely related to MS probands [71]. High risk of an MS diagnosis among closely related individuals suggests a link between HLA haplotype and the disease. There is strong support from multiple studies including large genome wide association studies (GWAS) for the HLA allele DRB1*15:01 as the most significant genetic risk factor for MS, particularly for a lowered age of onset [72]. This allele is also present at very high frequency (14%) in Northern European populations [73] and 18–19% in Southern European populations [74]. Its methylation status and co-morbidity with various environmental risk factors for MS are also well described as increasing MS risk [75,76].

There have also been at least a further 11 HLA alleles statistically associated with MS risk. A number of large GWAS studies (some including up to 80,000 individuals) have further identified up to 200 non-HLA genetic variants statistically associated with MS risk. These variants are, however, still only calculated to explain about 30% of the genetic risk of MS. It is also worth noting in this respect that many of these studies have been restricted to European ethnicity and that many of these genes have limited support for association and await functional characterization for a contribution to MS phenotype. There have also been a number of studies of rarer gene variants in specific families with high MS risk, some of which have been controversial [77]. The diagnostic confusion between MS and several rare gene disorders that can have similar presentations is also potentially a confounder in some of these studies [78,79].

## 2. Environmental Risk Factors for MS

### 2.1. Latitude

A connection between the latitude of habitation and the risk of developing MS has been extensively documented. For example, the majority of MS cases are registered in temperate regions where sunlight is less intense [80,81]. It appears that the risk of MS diagnosis increases with distance from the equator [82]. Even more so, higher numbers of MS cases are reported within the same country in higher latitude regions than closer to the equator [63,83,84]. It has been suggested that both skin color and UV exposure may contribute to the connection between the onset of MS and the latitude. An increased latitudinal gradient of MS prevalence is well documented, where the highest incidence rate is registered among patients residing above the 42° latitude [80,85] (Figure 3). Therefore, the high prevalence of MS in Scotland and England fits the latitudinal risk of the disease [86,87]. Strong evidence of the association between latitude and MS prevalence was presented in a meta-analysis by Simpson et al. [82]. Interestingly, the authors suggest that this association could be modified by local genetic and behavioral-cultural variations.

There is limited direct evidence to explain the role of the latitude in pathophysiology of MS. The most plausible explanation of why more MS cases are diagnosed at higher latitude is less sunlight exposure. Specifically, lower vitD production in those with less sun exposure has been suggested as an explanation for the latitude-associated differences in MS risk [89]. It has been proposed that the intensity of ultraviolet B (UVB) radiation, which is essential for vitD synthesis in the skin [90], is lower at northern latitudes [91]. This is because UVB light is readily absorbed by the ozone layer [92]; therefore the increase in the ozone path length at higher latitudes results in decreased UVB light reaching the Earth [93]. This explains low vitD production in all individuals living at higher than 33° latitude [94]. Similarly, no vitD synthesis was shown in individuals living at 90° latitude and none was found during the winter at latitudes above 50° (Antarctica, most of Greenland and Alaska, northern parts of Canada, Russia and Europe) [95]. These authors demonstrated that little vitD production occurs outside of the summer in all of these northerly places. Also, a significant decrease in vitD synthesis was detected in individuals that relocated from southern regions.

Interestingly, it appears that sunlight also has a direct immunosuppressive effect [96]. Using experimental autoimmune encephalomyelitis (EAE), an animal model of MS, Wang et al. have shown that UV radiation can ameliorate symptoms independent of vitD production [97]. Similar results were presented by Becklund et al. [98], where UV radiation inhibited inflammation and demyelination of the spinal cord. Also, UV radiation dramatically reduced CCL5 mRNA and protein synthesis as well as increased macrophage migration and IFN-γ production in the spinal cord. The authors suggest that UV radiation prevents the migration of inflammatory cells into the CNS by focal chemokine inhibition and a systemic increase of immunosuppressive IL-10.

### 2.2. vitD

Decreased serum levels of vitD in MS cases were demonstrated in 1994 by Nieves et al. [99]. Since then, the role of vitD in the pathogenesis of MS has been well established, where a number of studies provide strong evidence for correlation between low vitD level and the risk of MS [100,101,102,103]. Although a safe vitD level was not identified, it appears that 63.3–75 nM/L serum level of vitD is protective, as a lower risk of MS was found in these individuals [104,105]. This assumption is supported by the report made by the Institute of Medicine (IOM) Committee to Review Dietary Reference Intakes for vitD and Calcium in 2011 regarding the dietary requirements for vitD [106]. According to this report, a serum level of vitD below 50 nM/L could lead to serious health consequences. Also, according to the guidelines of the IOM and the Endocrine society, the “vitD sufficiency threshold” is 75 nM/L, which also correlates with stable serum parathyroid hormone levels [58]. Unfortunately, the majority of population around the world has levels of vitD below the threshold [58,107].

The serum level of vitD also changes depending of the month of the year, being lower during the winter and increasing in the summer in temperate regions, suggesting links to sun exposure. In another study, seasonal changes in vitD were found in the serum from a large American cohort, peaking in late summer [108]. Interestingly, changes in vitD differ between sexes, where vitD levels are lower in woman when compared to men [109]. More so, summer and autumn levels of vitD in women were significantly lower as compared to those in men. This could be related to the higher frequency of MS diagnosis in woman as compared to men [55,56,57]. Interestingly, vitD levels during pregnancy could be a risk factor for MS diagnosis in the child. Studies have shown that the month of birth correlates with MS risk, with a significant increase in MS risk among individuals born in April and May as compared to those born in October and November [110,111]. This observation indicates the association between maternal sunlight exposure and the vitD status of the mother during pregnancy and vitD deficiency of the fetus.

vitD replacement therapy has demonstrated an improvement in the mental health and decreased annual relapse rate in treated MS patients in several medium-sized clinical trials [112,113]. A large data met-analysis revealed a trend towards fewer relapses and improvement in expanded disability status scale EDSS for analyzed studies [114].

In addition to its well-documented role in calcium metabolism it is now clear that vitD also plays a significant role in immune signaling. The receptor for the activated form of vitD (calcitriol), (nVDR) is present in many cell types including monocytes, macrophages and lymphocytes, and functions as a nuclear ligand binding domain, regulating transcription of many different genes. The primary immune-modulatory effects of the calcitriol/nVDR complex are via modification of activation, differentiation and proliferation of immune cells. Type I pro-inflammatory cytokines such as IFN-γ are downregulated and type II anti-inflammatory cytokines (such as IL-10) are upregulated. The generation and activation of Treg and tolerogenic DC’s also occur in response to calcitriol signaling. The overall effect of calcitriol signaling is a shift away from inflammatory immune responses. vitD also directly modulates the function of brain pericytes (which maintain the blood brain barrier, BBB) by downregulating inflammatory responses in these cells [115].

Clearly calcitriol-mediated immune modulation is in the opposite direction to that which characterizes inflammation in MS disease. The production of vitD in response to sunlight exposure also clearly links to the latitude of habitation (which determines UVB exposure). These two risk factors are intrinsically tied together and provide an obvious reason for seasonal variation in MS relapses and the month of birth as an MS risk factor [116].

### 2.3. Lifestyle Factors

There have also been a number of studies linking lifestyle factors such as obesity, diet, changes in the gut microbiome, smoking, exposure to industrial chemicals such as organic solvents and an “urban” lifestyle with an increased risk of developing MS or of an increased risk of disease progression [117,118,119,120,121,122]. Much of the data for dietary and microbiome effects are inconsistent and there is little clear mechanistic explanation in many cases for how these factors are specifically linked to MS beyond the fact that poor diet, obesity, smoking and industrial chemical exposure are well recognized as inductive of a general pro-inflammatory state that could exacerbate other MS risk factors [123,124]

### 2.4. Viruses and Other Infecitous Agents

Environmental factors including viruses, where Epstein Barr Virus (EBV) infection and endogenous retrovirus reactivation have been connected to MS. EBV is a gamma herpesvirus which has an almost ubiquitous prevalence in the adult population worldwide with more than 90% of adults demonstrating seroconversion to the virus. It is spread by saliva and other secretions and primarily replicates in B cells, triggering their proliferation. The accepted pathogenesis of EBV-induced disease is that those infected when young (prepubertal) do not show overt disease, whereas those infected as adolescents or adults will display infectious mononucleosis (IM or Glandular Fever), involving swollen lymph nodes, malaise and fever due to CD8 T cell responses. The reason for this age-related difference in pathogenesis is not clear but is probably related to the fact that there is a general switch from more innate immune system driven to more adaptive immune system responses with age. Like many herpesviruses, EBV establishes latency in its target cells. Several proteins produced by the virus during latency that manipulate the host cells’ life cycle can also trigger B cell lymphomas. This effect is more marked in immunosuppressed patients such as those with AIDS [125,126].

The epidemiological evidence for the involvement of EBV in MS centers around studies that have demonstrated that the risk of MS is very much higher in those individuals that have suffered from IM than those that have not (2–3-fold higher risk) [125]; conversely, the risk of MS for those who are not seropositive for EBV (admittedly a small cohort) is 15 times lower than those that have had the virus [126]. In addition, antibodies and T cells responding to lytic phase proteins of EBV, indicating recent infection or re-activation of EBV, are also associated with MS.

Like all viruses, EBV triggers Th1 cell-mediated immune responses and like many herpesviruses it also encodes proteins that manipulate the host’s immune response for the virus advantage. These include proteins that inhibit the production of antiviral cytokines such as IFNα and IFNγ and proteins that interfere with HLA loading of viral antigens and proteins that inhibit apoptosis of immune cells [127]. It could be suggested that the type of antiviral immune response triggered by EBV correlates with that seen in MS pathogenesis.

Exactly how EBV infection (which occurs in the peripheral circulation) translates into CNS immune dysfunction in MS is not clear; current theories include leakage of EBV-infected B cells across the blood brain barrier triggering a pro-inflammatory environment, cross activation of antibodies against EBV towards myelin antigens, or dysregulation of the immune system due to EBV infection and manipulation of B cells and monocytes causing stimulation of auto-reactive T cells [128,129].

Conversely, seropositivity to Human Cytomegalovirus (CMV) a betaherpesvirus that also has a near ubiquitous prevalence in the adult population conveys a protective effect on MS risk [130,131]. However, this theory is clouded by frequent reports of CMV recrudescence in MS patients receiving immunosuppressive therapy [132,133] and that EAE susceptibility in adult mice can be induced by Murine Cytomegalvirus [133]. This confusing picture of pathogenesis, similar to that of EBV is most probably explained by the hygiene hypothesis, whereby exposure to generic infections early in life enhances the development of an immunoregulatory system that decreases autoreactive T cell activity later in life. Further evidence supporting this theory comes from studies of other pathogens that are normally acquired in childhood in poor hygiene environments such as *Helicobacter pylori* [134] and parasites such as *Toxoplasma gondii* [135] and *Trichuris trichuria* exposure, which also displays a negative association with MS. Though in the case of helminth parasites, there may also be direct immunosuppressive effects of excretory proteins from the worms that are being explored as potential MS therapies [136].

### 2.5. Endogenous Retrovirus Expression

A further viral risk factor is that of endogenous retroviral expression. Endogenous retroviruses are repetitive genomic sequences that are present in most, if not all vertebrates. They are thought to arise via infection of germ line cells via infectious retroviruses, as the lifecycle of retroviruses includes copying themselves into the host cells genome. The ones in the human genome are no longer active as infectious viruses but they are expressed both as RNA and proteins, and the regulatory elements in the longterm repeats at either end of the viruses interact with a variety of cellular transcription factors. The link between ERV expression and MS has been controversial; however, recent systematic reviews and meta-analysis of a large number of studies have clearly established that MS patients over-express RNA from the HERV-W ERV family when compared with healthy controls [137]. Other ERV families the expression of which has been associated with MS include: HERV-H, and in particular an SNP on one member of the HERV-H group, HERVFc1 on the X chromosome and HERV-K (particularly the polymorphic alleles HERV-K113 and HERV-K18 which are absent in some individuals) though the evidence for these is not as strong as for the HERV-W group. Single reports have also reported a link between MS and HRES and HERV-15 elements [137]. In terms of a plausible molecular basis for HERV-W in MS pathogenesis, expression of *env* proteins from this group of ERVs induces pro-inflammatory cytokines (such as TNFα) and can replace the adjuvant (typically Freunds adjuvant) used in triggering EAE in mice. Interestingly, EBV infection of B cells reliably triggers the expression of HERV-W loci by binding of the EBV gp350 protein. HERV-W expression is also increased in IM (EBV affected) patients, tying together these two risk factors in MS disease [138]. HERV-W is also constitutively expressed in the central nervous system and is crucial in placental function. The HERV-W member syncytin-1 is responsible for trophoblast fusion in the formation of the syncytia in human placentas [139]. It is feasible that immune responses against HERV-W, triggered by over-expression in EBV infection, could carry across the BBB into the CNS triggering local inflammation. The expression of HERV-W in pregnancy also provides a potential link with the role of steroid hormones (the expression of which is altered in pregnancy).

In a recent study, Manzari et al. showed an association of MS with a specific genetic variant of the nuclear antigen 2 (EBNA-2) [140]. These data provide strong evidence supporting the role of EBV in MS pathogenesis. Interestingly, EBNA2 directly interacts with the cellular DNA-binding protein RBPJ-κ (recombination signal-binding protein J kappa), a ubiquitous protein of the Notch signaling pathway that plays an important role in MS pathogenesis. RBPJ was suggested as a possible autoantigen in MS as it has been demonstrated to bind to CSF-derived IgG in some MS patients [141]. LMP1, another EBV protein, can induce expression of immunoglobulin kappa light chains [142], which are also consistently found in MS cases [143]. One multicentre study has shown the diagnostic value of these kappa-free light chains in CSF in diagnosis of MS [144].

## 3. Conclusions

Tying together such seemingly disparate risk factors in MS disease into a coherent model of disease pathogenesis is not straightforward, particularly as there are many missing pieces of basic knowledge about the molecular interactions of these factors with the immune system. However, a very basic outline is beginning to emerge. This could be best summarized as a pro-inflammatory environment consisting of a combination of low vitD levels (linked to low sunlight exposure), active EBV infection (particularly IM) and HERV-W expression creating the conditions for a TH17 driven autoimmune inflammatory response that targets myelin in the CNS. The role of steroid hormones (vitD and sex hormones) is clearly a protective one and provides an obvious explanation for the increase prevalence of MS in women compared with men. However, MS does not occur in all women with low vitD levels and EBV infection or only in people with these risk factors. There are also clearly underlying genetic predispositions, both at an individual and ethnic group level that make a crucial difference in the development of disease. It appears that the pool of lymphocytes reactive to myelin protein become developed outside the CNS and, eventually, cross the BBB and attach to brain neurons (Figure 4). It remain unknown which factors play a leading role in the generation of autoreactive leukocytes.

## Figures and Tables

**Figure 1 ijms-20-00303-f001:**
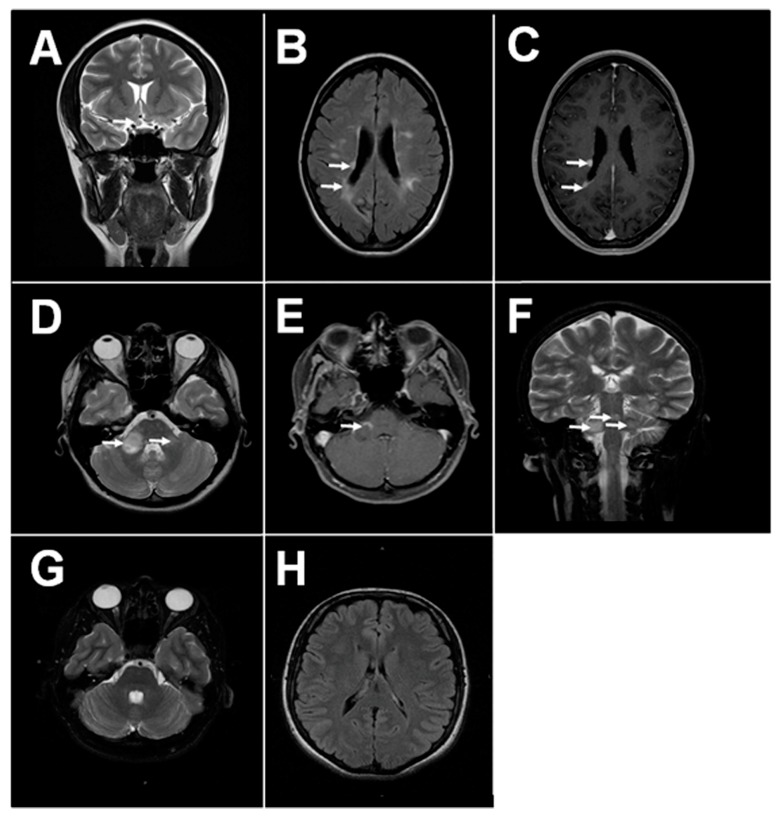
Magnetic resonance imaging (MRI) of MS and health brain. (**A**) Typical MS lesions in the right optical nerve on coronal T2 image; (**B**,**C**) Typical MS lesions in periventricular white matter on axials T2 and T1 post contrast images, respectively; (**D**–**F**) Typical MS lesions in the white matter of sub-tentorial structures (pons, right middle cerebellar peduncle): on axials T2 and T1-post-contrast, and coronal T2 images, respectively; (**G**,**H**) MRI of the healthy adult brain: normal sub-tentorial structures on axial T2 images, normal periventricular white matter on axial FLAIR images, respectively.

**Figure 2 ijms-20-00303-f002:**
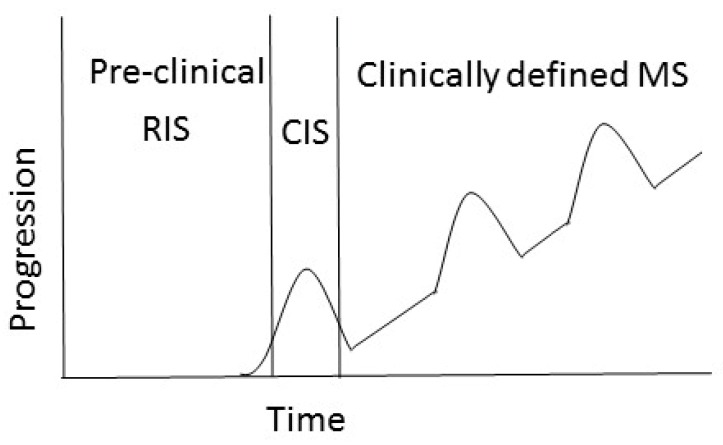
Clinical course of MS. The course of MS can be divided into clinically isolated syndrome (CIS) and clinically defined RPMS. Radiologically isolated syndrome (RIS) is defined by incidental imaging findings suggesting inflammatory demyelination in the absence of clinical signs or symptoms, while CIS is identified as the first clinical presentation of the disease. SPMS is diagnosed retrospectively by a history of gradual worsening after an initial relapsing disease course [53].

**Figure 3 ijms-20-00303-f003:**
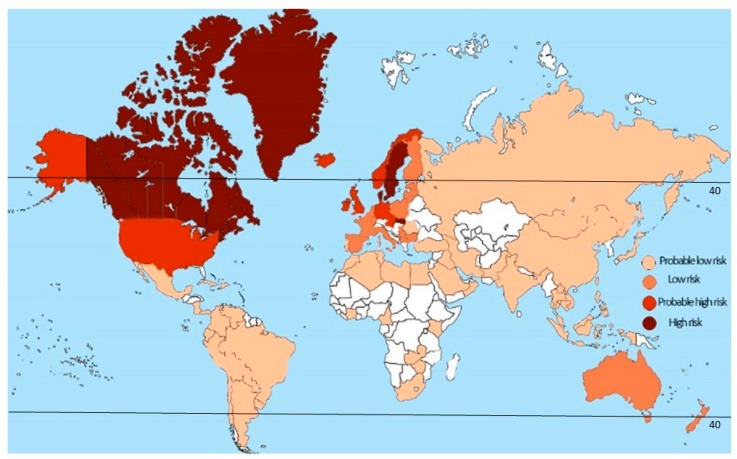
Global risk of development of MS. The prevalence of MS increases further from the equator in either hemisphere. Prevalence is higher in North America and Europe (291 and 232 per 100,000 respectively) and lowest in South America, Sub-Saharan Africa and East Asia, at 58 per 100,000 respectively [88]. It appears that highest incidence rate is registered among patients residing above the 42° latitude.

**Figure 4 ijms-20-00303-f004:**
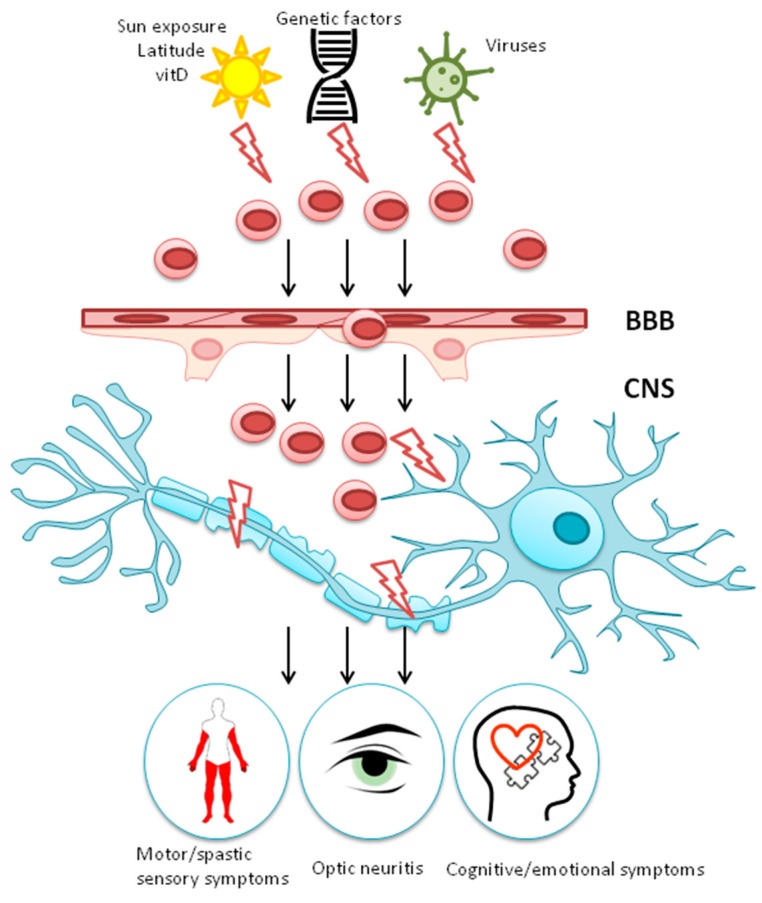
Pathogenesis of MS.

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
