# Peer review of "The Interaction between Viral and Environmental Risk Factors in the Pathogenesis of Multiple Sclerosis"

_ijms, 2019, doi:10.3390/ijms20020303_

Round 1
Reviewer 1 Report
This reviewer appreciates the additional improvements inserted the manuscript. The map figure as represented in the revised version still does not show the suggested changes to adhere to the data of the Atlas of MS from the WHO and the MS International Federation. In its present form the figure remains as outdated information which would not be adequate for the international audience of the Journal.
Author Response
We would like to resubmit our manuscript entitled “Environmental Risk Factors in Pathogenesis of Multiple Sclerosis”, the manuscript ID ijms-406757. Mild revisions were made according to the reviewer’s comments.
Please, find point-by-point revisions made in the manuscript:
Comments and Suggestions for Authors
1) This reviewer appreciates the additional improvements inserted the manuscript. The map figure as represented in the revised version still does not show the suggested changes to adhere to the data of the Atlas of MS from the WHO and the MS International Federation. In its present form the figure remains as outdated information which would not be adequate for the international audience of the Journal.
Respose 1: Agree, changes were made in Figure 2 to adhere to the data of the Atlas of MS from the WHO and the MS International Federation (https://www.msif.org/about-us/who-we-are-and-what-we-do/advocacy/atlas/).
Also, changes were made in the Figure legend: Figure 3. Global risk of development of MS. The prevalence of MS increases further from the equator in either hemisphere. Prevalence is higher in North America and Europe (240 and 192 per 100,000 respectively) and lowest in South America, Sub-Saharan Africa and East Asia, at 48 per 100,000 respectively [93]. It appears that highest incidence rate is registered among patients residing above the 420 latitude.
Sincerely,
Svetlana Khaiboullina, M.D., Ph.D.
Reviewer 2 Report
Line 36- Should be corrected...."the blood-cerebrospinal (CSF) barrier"I think
Author Response
We would like to resubmit our manuscript entitled “Environmental Risk Factors in Pathogenesis of Multiple Sclerosis”, the manuscript ID ijms-406757. Mild revisions were made according to the reviewer’s comments.
Please, find point-by-point revisions made in the manuscript:
Comments and Suggestions for Authors
1) Line 36- Should be corrected...."the blood-cerebrospinal (CSF) barrier"I think
Response 2: Agree, changes were made in the manuscript : the blood-cerebrospinal (CSF) barrier…..”
Sincerely,
Svetlana Khaiboullina, M.D., Ph.D.
This manuscript is a resubmission of an earlier submission. The following is a list of the peer review reports and author responses from that submission.
Round 1
Reviewer 1 Report
General comments:
This review mainly addresses specific virus infections as potential risk factors for MS. The title should be amended because the manuscript is poor in discussing other environmental/life style risk factors (except latitude and UVB/vitamin D). For example, the roles of diet, microbiome, organic solvents and urbanization are completely omitted. In what concerns infectious agents, the paper should include other players (such as cytomegalovirus and helicobacter pylori) which may have protective roles in MS pathogenesis (the hygiene hypothesis is not discussed). The interaction between environmental factors (including infectious agents) and the individual genetic background is of paramount importance and is not emphasized (in particular the interaction with HLA haplotypes).
Specific comments:
Line 36: choroid plexus is a structure of the Blood-CSF barrier, not of the BBB
Line 68: myelin basic protein is not… “the main constituent of myelin”…
Line 104: Gender is not an environmental factor!
Line 109: RRMS instead RPMS
Lines 306-307: the last sentence is not clear
Fig 1: This figure should be amended. Motor/ Somatic sensory symptoms instead of Spastic gait and paraplegia; and Cognitive/emotional symptoms instead of Muscle weakness
Fig 4: This figure should be removed from the Conclusion section and included in the Introduction (or removed from the manuscript)
Reviewer 2 Report
The topic you address is of substantial actual importance and interest. There are some observations from this reviewer:
(1) Figure 1 is too simplistic and mentions gene factors although these really do not form part of the environmental gamut of factors contributing to MS pathogenesis, unless they are confounded with the important field of Epigenetics which is not even mentioned in the manuscript.
(2) In page 2, second paragraph, the essential interactioon between cell-surface receptors i.e. VLA4 and VCAM are not mentioned.
(3) Figure 2 is outdated. Globally, the most utilized version is the proposal emited in 2014: Lublin FD, Reingold SC, Cohen FA, et al. Defining the clinical course of multiple sclerosis. Neurology 2014; 83: 278-286.
(4) Your discussion on prevalence related to altitude misses the most consequential factor in geographic MS distribution: genetics. More than 60% of peoples living in the hifgh risk areas, carry the most common genetic signature: HLA-DRB1*1501. In this sense, Figure 3 is not accurate. The map exhibits several areas affected by middle and high risk for MS, i.e. Uruguay, Spain, Middle East countries. Most accurate mapping in this regard are available from the Atlas of the MS International Federation and the World Health Organization (2013).